# The delivery of creative socially prescribed activities for people with serious mental health needs during lockdown: Learning about remote, digital and hybrid delivery

**Lindsey Bishop-Edwards**\* , **Elizabeth Taylor Buck** , **Scott Weich**

Sheffield Centre for Health and Related Research, Division of Population Health, School of Medicine and Population Health, University of Sheffield, Sheffield, United Kingdom

☯ These authors contributed equally to this work.

\* l.j.bishop-edwards@sheffield.ac.uk

**Data Availability Statement:** Data cannot be shared publicly because of confidentiality. Data are available from the University of Sheffield Ethics

## Abstract

### Background

Social prescribing interventions connect mental health service users to community resources, to support physical and mental wellbeing and promote recovery. COVID-19 restrictions impacted the delivery of socially prescribed activities, preventing face to face contact for long periods.

### Aims

The aim of this study was to understand how Voluntary Community and Social Enterprise (VCSE) organisations working with a local NHS mental health Trust responded to the challenges of social distancing during the COVID-19 pandemic. This understanding will be used to make recommendations for future practice, post-lockdown.

### Methods

Using a convergent mixed methods design, we surveyed VCSE providers of socially prescribed activities intended to be accessible and appropriate for people with severe mental health needs. Follow-up interviews explored further how they adapted during the first year of the pandemic, the challenges they faced, and how they sought to overcome them. The survey and interview data were analysed separately and then compared to identify convergent and divergent findings.

### Results

Twenty VCSE representatives completed the survey which provided a snapshot of changes in levels of connection and numbers reached during lockdown. Of 20 survey respondents, 11 participated in follow-up interviews. Interviews revealed that lockdown necessitated rapid change and responsive adaptation; activities were limited by resource, funding,

Committee (contact via scharr-rec@sheffield.ac.uk) for researchers who meet the criteria for access to confidential data.

**Funding:** This report is independent research funded by the National Institute for Health and Care Research Yorkshire and Humber Applied Research Collaboration. Award number NIHR200166. The views expressed in this publication are those of the authors and not necessarily those of the National Institute for Health and Care Research or the Department of Health and Social Care. The National Institute for Health and Care Research Yorkshire and Humber Applied Research Collaboration is a programme of research awarded to a group of academic Theme Leads in different institutions. Details can be found at: https://www.arc-yh.nihr.ac. uk/. The funders had no role in study design, data collection and analysis, decision to publish, or preparation of the manuscript.

**Competing interests:** The authors have declared that no competing interests exist.

safeguarding and government restrictions; no single format suited all group members; connection was key; and impact was difficult to gauge.

## Conclusions

VCSE organisations commissioned to deliver creative socially prescribed activities during the pandemic rapidly adapted their offer to comply with government restrictions. Responsive changes were made, and new knowledge and skills were gained. Drawing on experiences during lockdown, VCSE organisations should develop bespoke knowledge, skills and practices to engage service users in future hybrid delivery of arts, sports, cultural and creative community activities, and to ensure that digital activities offer an equivalent degree of connection to face-to-face ones. Additionally, more effective methods of gaining feedback about patient experience of hybrid delivery is needed.

## Introduction

The NHS Community Mental Health Framework for Adults and Older Adults emphasises the importance of services supporting people to participate in their communities [1]. A recent review of international community based social interventions for people with severe mental illness (SMI) found a growing evidence base for interventions designed to support community participation [2]. Benefits of engaging in community based social interactions include practicing and improving social interactions, increased confidence, reduced social isolation and a sense of purpose and belonging. Engagement with creative activities has also been reported to support wellbeing, help people feel happier and to promote faster recovery rates [3].

Social prescribing (SP) is one way that services link up health service provision with voluntary and community sector provision. The scope, purpose and motivation of SP is not clearly defined or agreed upon [4], however, in the context of mental health recovery, a focus on person-centred design and delivery and holistic approaches that provide an environment for individualised change and development, were found to be key [5].

South West Yorkshire Partnership NHS Foundation Trust (SWYPFT) developed an innovative bridging model to support and sustain community activities that are accessible and appropriate for people with severe mental health needs. The Trust has a funded charity embedded within its provision structure called *Creative Minds* (CM) (https://www. southwestyorkshire.nhs.uk/creative-minds/home/)

CM helps to fund and support a variety of local VCSE partner organisations to deliver community-based groups and activities such as choirs, art, gardening and football. These creative activities aim to "increase confidence, develop social skills and facilitate new experiences to improve the lives of local people" [6], with a particular focus on those who have mental health needs. This kind of connection has the potential to be transformative, providing purpose and meaning [7]. CM aims to connect SWYPFT mental health service users with community projects to promote and support recovery and well-being. Access to a network of partners with their varied range of group activities is intended to provide an opportunity for people to engage flexibly and creatively in alignment with their own individual preferences [3] with the aim of overcoming isolation and providing connection for people, including those considered harder to reach.

Creative Minds fits into the broad social prescribing model, though their work with secondary care services differs from primary care social prescribing. It is a less prescriptive process and many partners do not refer to their activities as social prescribing.

Organisations that CM partner with have traditionally offered regular face to face activities aimed at supporting recovery through shared interests and social connection. However, in March 2021, England was placed into lockdown due to COVID-19, and this included suspension of all group gatherings. The VCSE organisations partnered with CM had commitments to service users and funders and therefore had to find new ways to deliver their activities. The aim of this study was to understand VSCE providers' range of experiences of adapting to lockdown restrictions and to understand what they perceived to be beneficial to service users to inform delivery, post-pandemic.

## Materials and methods

### Study design

We used a mixed method design comprised of a structured questionnaire, followed by in depth semi structured interviews.

### Setting

Surveys were carried out at the end of 2020 with interviews being done over the first 3 months of 2021. Participants were asked to reflect on the first period of lockdown. They also had the additional awareness of subsequent lockdown and the need to plan for operating in person with restrictions. They also were able to reflect on hybrid working.

### Ethics statement

Ethical approval was sought and granted from The University of Sheffield Research Ethics Committee, reference number 034339.

The author(s) declared no potential conflicts of interest with respect to the research, authorship, and/or publication of this article.

Survey participants completed an online consent form before starting the survey. Interview participants were asked to read and complete an emailed copy of the consent form. This was returned by email to the researcher. If a participant was not able to do this verbal consent was taken at the start of the interview and recorded as part of the transcription.

### Participants

Participants were representatives from VCSE organisations working in partnership with CM to deliver creative community based activities intended to be accessible and appropriate for people with severe mental health needs. Organisations were eligible to participate if they had received support from CM in one of their recent funding rounds. An invitation to participate in an on-line survey was circulated by CM to all their eligible partners. Survey respondents were asked whether they would be interested in taking part in a more in-depth discussion with a researcher. All those who expressed an interest were contacted to arrange a semi-structured interview. Our aim was to get as many survey responses as possible and to conduct 8–10 interviews or as many as were needed to achieve theoretical saturation. The sample size achieved for the survey was to be all responders, which was 20 for the survey and 11 for the interviews, with an aim for 8–10 interviews to be secured from volunteers from the survey. 11 volunteers were identified, therefore interviews were increased to 11 for completeness. Responses from 20

VCSE groups represents approximately 40% of the amount of groups that were being supported by CM at the time of lockdown.

Across the 20 survey respondents, and across the interview group, the activity range was, broadly, arts and literature, horticulture and social care support. Of the 11 interviewees, 10 were female and one was male.

We used a convergent mixed methods study design in which the survey data and interview data were analysed separately, and then reviewed together to gain a more complete picture of how social prescribing providers responded to lockdown, the challenges faced and the solutions they found. The survey and interview data were compared, and it was noted where the findings of the survey and the interviews converged or diverged.

## Materials

The survey was drafted by two researchers (LBE and ETB) before being shared with CM staff, SWYPFT research staff, and a service user lived experience advisory panel (LEAP). The LEAP are a group of people who have lived experience of mental health services and who have an interest in research and in supporting projects to be both relevant and ethical. Participants suggested where language and content could be improved and made more understandable and accessible. The survey questions included a mix of fixed option answers, fixed parameter numerical answers, and free text answers. There was no limit on the size of response, but the answers tended to be of moderate to short length. The survey was built in Qualtrics, and a unique code generated to allow participants to access it.

The topic guide for the follow-up semi-structured interviews was based on responses to the survey questions, with prompts added to explore areas that were of specific interest to the research question. The topic guide was reviewed by the LEAP members and amendments were made as a result.

## Procedures

A description of the research and an invitation to take part was sent by CM staff to their partner VCSE organisations. Responses to the survey were sent directly to the research team and CM had no further involvement in the recruitment or data collection. The survey included an information page about the research and a consent form which, once completed, led the participant to the survey questions. Questions could be missed out if the participant chose to do so. CM sent out one reminder to their partners to invite them complete the survey.

All survey respondents who expressed an interest in taking part in an interview were contacted. Two follow up contacts were undertaken to maximise chances of recruitment whilst respecting the range of difficult circumstances many people were in during lockdown. Because the study took place during lockdown, interviews were conducted using Google Meet. Cameras and video functionality was switched off in accordance with the ethical approval given. Two researchers conducted the interviews (LBE and BD). At the start of each interview the researcher took consent to ensure that the participant was happy to proceed. The researcher asked questions from the topic guide and used the prompt questions to gain a richer understanding when appropriate. During the interviews participants were asked to describe: the activities they provided prior to lockdown; how they adapted or changed these activities during lockdown; which on-line platforms they used; how they stayed connected with people who did not like using on-line platforms; the challenges encountered; and which practices they intended to continue using.

### Analysis

**Quantitative and qualitative analysis of survey data.**   The quantitative and qualitative (free text answers) were reviewed and collated as a combined data set. Quantitative data about numbers of participants lost or gained were checked against corresponding descriptive text to ensure the numbers given were actually aligned with the written descriptions given by the participants. If answers were unclear and respondents were available for interview, then further clarification was sought to ensure accuracy of reporting. Survey respondents were split into four groups to enable comparison between them. These groups were based on the number of additional members reached during lockdown. The data were analysed to understand the difference between providers that had not expanded their reach, those who had done so moderately, and those who had expanded to a greater degree. This comparison allowed patterns to be seen across the data.

Results of quantitative and qualitative analyses were triangulated to obtain a richer, more complete idea of how providers adapted during lockdown and their perception of the challenges they faced. The findings from both parts of the study were compared and the research team identified where these converged, were confirmatory, and where there were discrepancies or differences.

### Qualitative analysis of interview data

Data from semi-structured interviews were analysed thematically using an inductive approach [8] by the lead researcher (LBE). Familiarisation was followed by descriptive coding of the transcripts and the development of a coding framework from which themes could be developed. A random sample of transcripts was coded again by the second author (ETB) for critical discussion and development of the codes and themes. Both researchers independently and coactively grouped codes into themes that were pertinent to the research questions. Themes were then labelled and refined through individual reflection and joint discussion. These themes were also presented to the LEAP for reflection and review. The researchers selected representative quotes from the data for each theme.

## Results

### Survey findings

Responses were received from 20 VCSE organisations. Two VSCE providers of creative socially prescribed groups suspended their groups with no alternative creative activity offered. However, both offered some form of individual contact to members throughout lockdown. Two other providers reported that they had managed to continue with some face-to-face activities because these took place outdoors and were sufficiently small to not be impacted by restrictions. Five providers were able to offer some face-to-face contact, such as doorstep visits, because they were one-to-one and carried out within social distancing guidelines.

Nine providers offered regular remote group activities, while twelve offered one-to-one contact via text or telephone. Sixteen of the 20 respondents said they had provided materials for contact-free activities. These findings are summarised in Fig 1. Sixteen of respondents reported that they had moved to some sort of digital online provision or engagement using Facebook, Zoom, YouTube and Teams. Some respondents highlighted that it was particularly difficult to engage older adults.

The 16 groups that had created an online presence stated they gained new members and expanded the reach of their organisation during lockdown. Some respondents reported that numbers had increased by over 80 while others reported a more moderate increase (see Fig 2).

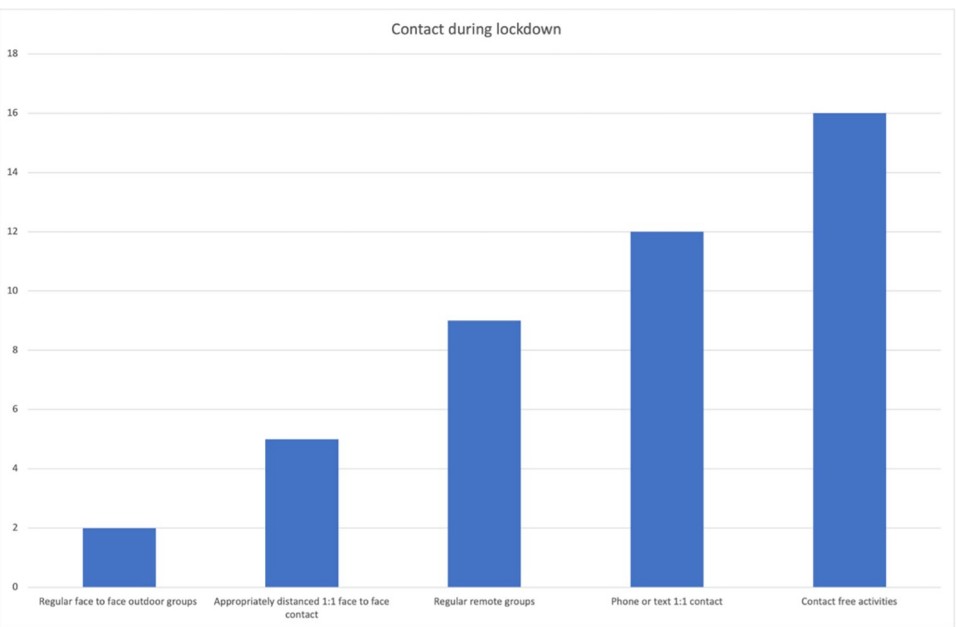

**Fig 1. Level of contact provided during lockdown.**

People accessing activities and social media during lockdown were not necessarily the same people who accessed the group prior to the pandemic.

Respondents reporting no increase in numbers tended to have either suspended their groups entirely, or to have taken their pre-pandemic activity online with little other adaptation to the original format. They reported that they had decided to engage only with people who were already part of the group. One reason given for this was pressure from staff shortages combined with skill deficits, which impacted delivery. Respondents reporting moderate expansion tended to report some use of online group meetings but mainly to have set up private or closed social media groups. They described some degree of collaboration with other organisations such as cross-working and sharing of resources.

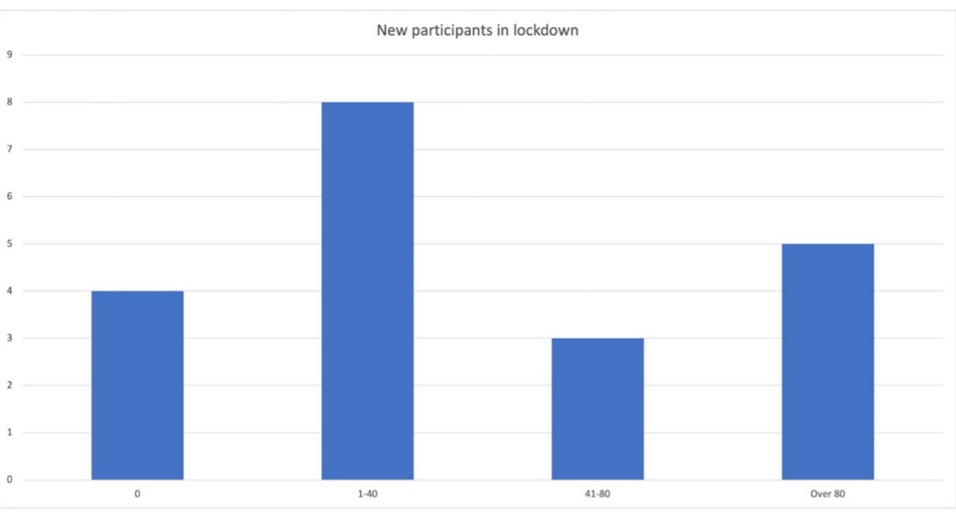

**Fig 2. New participants gained in lockdown.**

Respondents reporting greater expansion described distributing resources for activities via existing or new points of contact for example dropping materials off in residential settings or care homes. They used a mix of public and private social media spaces, making the distinction that sharing feedback was predominantly private, but activities were public and open to anyone. Respondents reporting the highest levels of expansion tended to have higher levels of collaboration with organisations they had previously worked with for example arts councils, local councils, or other partners. They also reported that they had used community advertising, networks, newsletters, and social media pages of their partners. They had set up more open access social media groups and had widened the types of activities offered. Respondents reporting the most expansion were also offering multiple modes of contact and engagement.

## Semi-structured interview findings

The sample for the qualitative analysis comprised representatives from 11 community organisations. All participants interviewed were in a co-ordinating or managerial role within the organisation that they worked.

Five main themes were identified from the qualitative data. These were: rapid and responsive adaptation; pushes and pulls shaping lockdown provision; one size does not fit all; connection as key; impact was difficult to gauge.

## Rapid and responsive adaptation

Lockdown and the embargo on group activities necessitated rapid change on the part of providers which led to new learning. Eight of the 11 VCSE representatives interviewed described how they had rapidly expanded their online presence and provided online alternatives to the face-to-face activities they had offered pre-pandemic. Several representatives described on-line provision as being outside their comfort zone, but something that they had either intended to do in the future, or something they now recognised as a valuable skill to take into their future provision. The learning they reported included improved technical skills in delivering online pre-recorded tutorials; tutor-led live sessions; and drop-in sessions.

*We've learnt, I mean all of us, our skills base has now really amplified.* (P2)

There was also learning in relation to online security and safeguarding, including the use of private Facebook groups and online breakout rooms.

*[O]n Zoom. . .. even though there are breakout spaces. . .that presents with it a lot of safeguarding responsibilities to us.* (P10)

Although use of social media resulted in some organisations reaching more people, they reported that they did not know much about the people receiving packs or activities. As these resources were freely available to the whole community, providers were unable to determine if people with serious mental health needs were accessing or using them.

Some groups did not need to move online to provide their activity, and these were predominantly those who had access to outdoor or large spaces, usually for gardening or food-based growing activities. These outdoor open spaces were seen as invaluable in the context of the pandemic due to being outside and therefore less constrained by coronavirus restrictions. However, even in these groups, COVID-19 restrictions still had an impact which required response and adaptation. Socialising before or after sessions and providing drinks had to be stopped, which was perceived as having a negative impact.

*[The] sociability part of it is, you know, was a big thing.* (P3)

All providers had to rapidly find ways to maintain connections with and between group members which meant doing things they would not have done before like calling, emailing or visiting members directly to see how they were. This was driven by a perceived need to maintain connection, and to check on members' wellbeing.

*[B]ecause of regular phone calls and like the doorstep stuff. . .we've been able to open up a bit of erm- open up more interaction.* P9

*[F]or example I had a really active member and then she just sort of disappeared. . .in the end I just phoned her up and said are you alright?* P2

Overall, providers reported that the learning they gained from the pandemic was a positive outcome from a difficult situation, and that a number of practices would be taken forward into everyday practice in the future:

*[T]hey've been so successful [online sessions], I think we are going to continue, not you know not having as much online presence. . .offering face to face activities but also as well replicating those online.* (P13)

## Pushes and pulls shaping lockdown provision

Respondents reported a number of pushes and pulls that shaped the changes they made to their groups and interactions during lockdown. There included practical and legal restrictions; funding and resources; and participants' preferences.

All respondents reported that practical and legal restrictions relating to lockdown shaped the changes they made to their provision. Lockdown and the 'stay at home' mandate shaped both what they delivered, and how they delivered it:

*As soon as we went into lockdown [on] 23$^{rd}$ March all our groups stopped and I thought . . . right well we'll just go in a different direction.* (P2)

To deliver new formats, most groups needed to find additional resources for their activities, such as new technology and additional craft materials, which were at times difficult to source. Paying for these resources was cited as a challenge. Some organisations used funding they already had, some applied for additional funding that had been made available due to the pandemic lockdown, and some used both. While there were some savings made on things like rental of group premises, groups generally found that they were financially stretched:

*[E]verything was twice as hard and half as good really. . .we needed more staff hours [and] there was more being spent on resources.* (P9)

Providers reported that any changes made to delivery formats were shaped by original funding bids, stipulations about additional funding, and government restrictions and guidelines:

*[A]t the moment the funders are yeah, they want, erm each product you make or each activity that you do they want to know that it's. . . got a range of access points.* (P8)

The preferences of the people they were working with also shaped provision and how decisions were made about what changes to make. The closure of schools and childcare facilities led to some providers diversifying their provision to appeal to wider audiences, for example activities that could be done with children:

*[D]uring the first lockdown obviously, people were trying to be creative with all sorts of things and it gave something which parents could do with kids.* (P21)

Changes were made to the times of sessions to increase accessibility. This meant that the restrictions affected how the tutors and group leaders worked as they adopted a more fluid and responsive working style to meet the needs of their group members and to increase accessibility:

*[A]ctually sort of being a bit more, being it's ok to be responsive at this time, rather than what were used to being able to do is plan, control and work towards clear end goals.* (P10)

Some providers who ran online groups actively reviewed the groups' needs through talking to people, texting and emailing. People's needs were seen to change over time which made a review, respond and adapt process important.

Some providers described the importance of 'holding a safe space' for their group members, trying to be present and available to help people who were now restricted from going to face to face groups. This 'safe space' was linked to well-being and mental health needs.

*[T]he most important aspect of what we've been doing and . . . that positive mental health, where actually just being there and . . . holding a safe space* (P10)

## One size doesn't fit all

The providers interviewed reported that, pre-pandemic, they provided a range of activities for a range of different participants. This included outdoor gardening, arts, and theatre groups accessed by a range of people including those with mental health needs, young people and neuro-divergent adults. Variation in their service user needs and preferences meant that, when activities moved on-line, each provider had to assess, adapt and tailor provision. Providers found out early on that there was no single format or solution that would suit all their participants:

*It's about recognising that. . . not one size fits all and doing as much as we can to reach out to as many as we can* (P13)

Most providers changed their offer at the start of the pandemic to a mix of individual-at-home activities and online provision, including online tutorials, Facebook pages and groups and Zoom meetings. Some reported that the changes they made to their delivery, increased accessibility and even allowed people to engage who previously might not have felt able:

*[They] would never, ever have come to anything that we would have done in our studio space. Too shy. . .they would never have come because they were too housebound, some of them are agoraphobic, completely lost their confidence.* (P2)

However, providers also reported that in some instances accessibility was reduced, either practically, or due to personal preference:

*[T]here was a Facebook live thing where people could comment on the side and interact, . . . what we found was . . . there was only a very limited number of people . . . who could partici-pate in this and that was because of the ability to either have the technology, and even if they had it, to use it.* (P9)

Providers cited the importance of understanding pandemic-specific barriers to engagement for their group members. This included: loss of daily structure; isolating for health reasons; lack of internet or technology; no time to themselves; inability to travel to outdoor activities due to the suspension of public transport; relapse; lack of physical contact and interaction; and not feeling safe in an online group. They described how they had tried to find ways to over-come some of these barriers including closed rather than open FB groups; informal calls and doorstep visits, teaching IT skills; setting up dedicated phone lines for people to talk; emailing or messaging people directly; practicing mask wearing; and doing practice Zoom sessions to build participant confidence.

Some providers decided early on that their group members would not respond well to online provision and therefore chose alternative activities such as wellbeing packs that mem-bers could work on at home:

*[A] lot of our audiences . . . come to us because they . . . want to be hands on, erm and a lot of them don't have the equipment, the technology. . . so this is why we came up with this as an idea [creative art packs] really as opposed to something online.* (P8)

Hybrid delivery was also seen as potentially beneficial and a useful tool to use post-pan-demic to increase accessibility and fit with the needs and preferences of more people:

*[I]t's kind of opened a lot of more access routes for everybody really.* (P13)

However, any potential future changes were seen as dependent on financial backing with providers needing convince funders of potential benefits.

## Connection is key

Providers reported being able to develop new connections during the pandemic, expanding their reach to community members who had not previously used their group. This was facili-tated by increased use of social media; offering online activities and activity packs to anyone who wanted them; and by using networks and relationships within the Trust, local councils, local authorities and with other VCSE organisations to maximise supplies, provision and advertising:

*[W]e linked with a local organisation. . .who were producing seed packs. . .I offered these to the families.* (P18)

Although some new connections were successfully established, providers also described instances of immediate and gradual disconnection with pre-existing group members. Immedi-ate disconnection occurred when the group did not move to any online provision or when pre-existing members did not feel comfortable, or were not able to, connect or share experi-ences online:

*[W]e did like online galleries so we . . . had done some of that and sharing stuff on social media but all of that, all of that only works for people who are interested in social media,*

*happy to have their stuff shared in a way where they have no idea who's going to look at it.* (P8)

Additionally, some people engaged with online formats at first, but these connections proved hard to maintain. Tutorials and online provision were reported to be well received initially, but over time some people experience online fatigue:

*[T]hat was the feedback we were getting really was that they...were wanting to get away from the computers and be around people.* (P8)

Providers reflected on what people were missing from the in-person group, and on their role as the provider, and concluded that in the past a key benefit of the groups had been a sense of connection and shared purpose. They reported that one of the key values of their group was to bring people together in a safe space where socialisation, a sense of belonging, shared experience, building relationships and connection could be enabled:

*The socialisation I think is one of the biggest things.* (P18)

*[W]e realised even though we loved to think it's the content that's the most important, I think at that particular moment in people's lives, it was the, the social aspect...what we have is quite unique which is about those relationships with the people. (*P10)

When providers identified that pre-existing group members had disengaged, they reported that they felt it was important to implement additional actions beyond the usual remit of their role. They used a range of personalised interventions to overcome the perceived barriers to engagement or to undertake regular checks on people's wellbeing. As discussed previously, these interventions included: teaching IT skills; practicing mask wearing; setting up dedicated phone lines for people to talk; messaging either via direct social media or phone; emailing; calling and visiting people.

Groups using outdoor spaces, who were able to continue with some degree of in-person contact, also reflected on the importance of the connection and the value of being able to continue to access a physical, and safe, space:

*We had some really nice feedback from them about how important it had been to be able to access the space during, especially during the full lockdown.* (P3)

Finally, even remotely, some level of connection and sharing of ideas, was also found to be possible:

*[Participants could] share their own activities that we could then . . . put out in the next pack to sort of say, "We've had this idea from so and so, from Joe Bloggs down the road" . . .. you know it was very much . . . sharing ideas.* (P13)

## Impact was difficult to gauge

Providers reported that pre-pandemic they would gather formal, informal, verbal and non-verbal feedback from group members during face-to-face activities, creating a continuous feedback loop. When the pandemic restrictions meant that the types of activities offered had to change, there were some examples of increased engagement and more visible and immediate evidence of impact:

*[T]hey say ooh I notice you've got a Zoom creative writing workshop going on, through [your] Facebook page, I would like to come to that with my friend. . . . They're all engaging aren't they, and they're liking, and they're thinking about it, and they're reacting and responding.* (P2)

However, more generally the COVID-19 restrictions were seen as reducing the feedback providers were able to get from their participants. Posting out activity packs and similar socially distanced initiatives did not generate the same level of feedback as pre-lockdown activities had and, even when formal requests for feedback were made, providers struggled to gather sufficient data to demonstrate impact:

*I emailed the group leaders for every group; I emailed them twice and I only had replies back from [one setting]; that was it! (P7)*

This meant it was not possible to gauge who the providers were reaching and the extent to which their activities were impacting on people's physical or mental wellbeing:

*[I]f we got feedback it was positive, if we didn't get feedback which mainly, we didn't, you know we didn't see the feedback, I wouldn't know. (P8)*

*[Do] they get that pack, and open that pack, and then actually do something? But we don't know about that because they then choose not to put that on the Facebook page. (P2)*

Critically, it meant that there were not many ways to ensure that they were engaging with and providing the right services for the most vulnerable and hard to reach people:

*We do have some people who they kind of drop in and out with our service and actually they're probably the ones that are most vulnerable. (p18)*

## Mixed methods synthesis

For the purposes of this comparison, we excluded survey data from organisations that were subsequently interviewed. This strengthens the findings because it triangulates the experience of the 11 interviewees with that of the 9 non-interviewees. The matrix in Table 1 shows the extent to which the interview findings align with the data collected in the survey. Two researchers (LBE and ETB) discussed how the themes from the interviews related to the survey findings until agreement was reached for each finding. The researchers challenged each other's decisions to ensure that full consideration had been made of each theme and the extent of the convergence/divergence.

## Additional findings from survey not found in interviews

The survey data highlighted staff shortages and a lack of skills or infrastructure impacted on what was offered:

*[N]ot having the skills, equipment or infrastructure to allow us to offer alternative opportunities. (P2)*

The survey also identified a further financial shortfall during the pandemic, i.e. the weekly payment made to the group by their members. These lack of funds from members impacted provision:

**Table 1. Matrix of findings from each component.**

| Theme | Finding from interviews | Survey convergence/ divergence/ not elicited through survey |
|---|---|---|
| Rapid and responsive adaptation | Move to online provision | Mostly convergent |
| | Rapid skill building needed | Some convergent |
| | New security and safeguarding skills needed | Not elicited through survey |
| | Outdoor groups adapted to comply with guidelines | No outdoor groups in non-interviewed sample |
| Restrictions shaping responsiveness | Practical restrictions shaped response | Some convergent |
| | Legal restrictions and guidelines shaped response | Mostly convergent |
| | Funding shaped response | Some convergent |
| | Practical needs of their group members shaped response | Mostly convergent |
| One size does not fit all | Variation in group member needs including their mental health needs | Mostly convergent |
| | Pandemic specific challenges identified | Some convergent |
| | Hybrid delivery seen as potentially helpful | Mostly convergent |
| Connection is key | New connections were made via social media | Mostly convergent |
| | Connections and preferences changed over time | Some convergent |
| | Reflection and learning about in-group connection took place | Some convergent |
| Impact was hard to gauge | Previous feedback loops and channels were reduced or lost | Some convergent |

*[M]embers usually pay a weekly sub for live workshop sessions.* P19

Survey data showed that some providers chose to focus only on those who engaged, which was linked to the lack of funds and perceived difficulties associated with moving their provision online for older adults

*It is hard to see how we can do much more in these times as we don't think the elderly will want to do anything like Zoom (P12)*

*We have only be able to offer limited services to specific groups. We have only engage[d] with those willing and interested in [participating] first remotely.* (P20)

Finally, the survey elicited a response that highlighted the shortcomings of resorting to sending out at-home activities:.

*[Sending out seeds and drawing equipment] is not getting them out of the house or meeting others.* (P12)

## Discussion

This paper explored how VCSE organisations commissioned to deliver creative socially prescribed activities for people with serious mental health needs were forced to rapidly adapt their offer to comply with the COVID-19 pandemic government restrictions. We found that creative groups' responses ranged from almost total shut down, to limited provision, to expansion to a hybrid range of offers including online, telephone contact, and at home activities. The changes made meant that learning and upskilling was rapid. This was further enhanced by listening to group member needs such as the need to feel safe online, enabling private online groups and areas, and safeguarding. These new skills were viewed as useful additions for the future for hybrid provision, though this would require funders to understand these benefits and to allow extra funds for groups to expand their offer.

The research showed that an increase in online provision, and the use of social media, led to an increase in engaged users, though it was difficult for providers to know who those new users were, what their individual needs were and the extent to which these needs were being met. Providers who might previously have seen the creative activity as the primary mechanism of change in supporting mental health, were reminded during lockdown that connection is also an essential pillar of recovery [9]. Being able to assess the level of connection for online users, in the same way that they might assess this verbally and non-verbally for in person attendees is therefore essential for future hybrid provision.

Additional benefits to hybrid provision of socially prescribed activities were that it did enable some people to join and connect during lockdown; a period where connection was inhibited. Post-lock this presents an opportunity for maintained connection for those who may come to experience non-Covid related isolation due to practical, physical, or mental health reasons. Examples might include people staying on hospital wards, relapse, ill health, childcare issues or people who struggle to leave their houses or meet others in in person.

We found evidence that, whilst some were enabled by online provision, such as those who would not have the access or confidence to attend in person groups, there were some groups that were less engaged, such as older adults. The range of needs encountered by the providers is significant. Recent research suggests that most people with serious mental health needs were limited users of the internet during the pandemic [10], falling on the wrong side of the pervasive social and information inequality known as the 'digital divide' [11]. Consequently, there is a risk that any future shift towards increased digital or online provision of socially prescribed activities will inadvertently increase social and health inequalities. The accessibility, acceptability and usability of digital provision for participants with serious mental health needs should be assessed as a matter of priority when planning hybrid groups and activities.

Before the pandemic, providers were unlikely to routinely gather data about digital competency, learning difficulties, additional needs, or personal preferences for remote, hybrid or face to face contact. This meant that an evaluation of how lockdown impacted on accessibility and uptake in the community as a whole, as well as for specific demographic groups, was not possible.

Social prescribing has the potential to address the social determinants of health inequalities [12]. To achieve this, provision must be accessible across the social gradient. Issues such as age, race, poverty, illness, and disability, which could increase isolation, should be at the forefront of providers minds, and demographic data on access and impact should be collected to ensure harder to reach groups are not excluded and to understand if provision is increasing or decreasing social inequalities.

## Strengths and limitations

This study was conducted in a single geographical region, with VCSE providers all linked to the same NHS mental Health Trust. That, alongside the small sample size should be taken into consideration when reflecting on the relevance of the results to a wider population. Local factors and idiosyncrasies will have impacted on participants experience; however, it is likely that there will be commonalities with people delivering and receiving socially prescribed activities nationally and internationally.

The data obtained in the survey was occasionally difficult to interpret due to the brevity of answers. It would have been beneficial to interview all respondents to clarify some of the comments. All respondents were invited to interview but not all respondents were able to do so.

This study explored the learning and experiences of VCSE providers during lockdown. One of the key limitations is that it does not directly report service users' views. A lived experience

study carried out in the same period [13] also found that creative activities provided a positive outlet and that use of technology was both positive and negative. However, further research is needed to gain a better understanding of service users' experiences.

## Conclusions

Prior to the COVID-19 pandemic, VCSE organisations provided an important sense of connection for people with serious mental health needs. Our findings show that VCSE organisations commissioned to deliver creative socially prescribed activities appropriate and accessible for people with serious mental health needs during the pandemic had to rapidly adapt their offer to comply with government restrictions while trying to meet the specific needs of their group members. New knowledge and skills were gained but connection with some people proved hard to maintain and evidence of impact was difficult to gather, leading to uncertainty about whether the most vulnerable or hardest to reach people benefited from what was offered. The impact on social and health inequalities was not formally evaluated by organisations delivering activities during lockdown.

These findings highlight that significant thought, planning and evaluation is needed to ensure that connections are maintained with people most at risk of exclusion when remote or digital creative groups or activities are planned or offered in the future. Understanding which aspects of lockdown provision were successful and which were not is imperative for future service developments. Robust feedback systems are needed to understand the delicate balance between maintaining and increasing reach and diluting interpersonal connection for the most vulnerable members of the community. It is also vital to understand and investigate further the ways in which connection and creative content interact to support recovery. Further research is needed to explore the lived experiences of members of the creative groups, including the benefits and challenges of both online and in-person creative socially prescribed activities.

## Acknowledgments

We would to thank all the organisations that took part in this study, both in the survey and the interviews at what was a very pressurised time for their organisations.

We would like to thank our LEAP advisory group who provided honest, insightful and valuable feedback throughout the research process.

We would also like to thank Brigitte Delaney for undertaking two of the interviews.

## Author Contributions

**Conceptualization:** Lindsey Bishop-Edwards.

**Data curation:** Lindsey Bishop-Edwards.

**Formal analysis:** Lindsey Bishop-Edwards, Elizabeth Taylor Buck.

**Methodology:** Lindsey Bishop-Edwards, Elizabeth Taylor Buck, Scott Weich.

**Supervision:** Scott Weich.

**Writing – original draft:** Lindsey Bishop-Edwards, Elizabeth Taylor Buck.

**Writing – review & editing:** Lindsey Bishop-Edwards, Elizabeth Taylor Buck, Scott Weich.

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
