## [Decision Letter · Decision Letter 0]

1 Aug 2023

PONE-D-22-21627The delivery of creative socially prescribed activities for people with serious mental health needs during lockdown: Learning about remote, digital and hybrid deliveryPLOS ONE

Dear Dr. Bishop-Edwards,

Thank you for submitting your manuscript to PLOS ONE. After careful consideration, we feel that it has merit but does not fully meet PLOS ONE’s publication criteria as it currently stands. Therefore, we invite you to submit a revised version of the manuscript that addresses the points raised during the review process.

We look forward to receiving your revised manuscript.

Kind regards,

Bronwyn Myers

Academic Editor

PLOS ONE

Journal Requirements:

Additional Editor Comments:

Thank you for your submission to PLOS One. We have now received two referee reports. based on their feedback, we will require major revisions to your manuscript before it can be further considered for publication. In particular, please focus on providing additional methodological detail, and critically contextualising this study and its findings within contemporary understandings of personal recovery and social prescribing for recovery from mental health concerns.

Reviewers' comments:

Reviewer's Responses to Questions

**Comments to the Author**

1. Is the manuscript technically sound, and do the data support the conclusions?

Reviewer #1: Yes

Reviewer #2: Partly

2. Has the statistical analysis been performed appropriately and rigorously? 

Reviewer #1: N/A

Reviewer #2: Yes

3. Have the authors made all data underlying the findings in their manuscript fully available?

Reviewer #1: Yes

Reviewer #2: Yes

4. Is the manuscript presented in an intelligible fashion and written in standard English?

Reviewer #1: Yes

Reviewer #2: Yes

5. Review Comments to the Author

Reviewer #1: The manuscript is a week written one with an exhaustive methodology. Kindly check the manuscript for some syntactical and grammatical errors before the final submission to consider ahead for publication.

Reviewer #2: Thank you for the opportunity to review this manuscript. The manuscript addresses the important topic of promoting continuity in recovery-focused social activities during the pandemic. There are several areas that could benefit from strengthening:

1. Abstract: The conclusions repeat the findings- in the conclusion please describe the main take-home message. liines 47-50 can be removed

2. Introduction. lines 82-84 are irrelevant and can be removed. Overall, the introduction could benefit from greater attention to the literature on personal recovery and the role of social prescribing. It is a bit thin and atheoretical. there is now plenty of literature on the impacts of covid-1i protective measures on mental health services that could also be referred to.

3. Methods. more information is needed on sample size (target), response rate, and a description of the participants. Please describe the LEAP and their composition and function. Please also provide more information on the quantitative analysis methods. Synthesis should be described under the Analysis subheading - it does not warrant its own subheading. Under synthesis please also describe how you created the amtrix reported in Table 1.

4. Results: The figures are redundant as most information is also reported in the text. Lines 225-227 should be reported in the analysis subsection.

5. Discussion: The key messages and implications of the findings are somewhat unclear. The discussion could benefit from highlighting these more explicitly. Please remove all subheadings from the discussion. The list of recommendations at the end of the manuscript should be removed and either integrated into the text, where these flow from the findings being discussed or removed entirely (where they do not directly arise from the findings being reported).

6. PLOS authors have the option to publish the peer review history of their article (what does this mean?). If published, this will include your full peer review and any attached files.

Reviewer #1: **Yes: **Pragya Lodha

Reviewer #2: No

---

## [Author Response · Author response to Decision Letter 0]

22 Sep 2023

Response to reviewers

Thank you for your helpful comments and guidance regarding the resubmission of this manuscript. We have incorporated the feedback and made changes as requested. Please find details of these changes documented below.

Reviewers’ comments to the author.

Reviewer #1: The manuscript is a well* written one with an exhaustive methodology. Kindly check the manuscript for some syntactical and grammatical errors before the final submission to consider ahead for publication.

This has been done throughout and if any further grammatical or syntactical errors are spotted we would be happy to make appropriate changes. 

* The original feedback said “week” which given the context of the rest of the sentence we have assumed is a typo that should have read “well”.

Reviewer #2: Thank you for the opportunity to review this manuscript. The manuscript addresses the important topic of promoting continuity in recovery-focused social activities during the pandemic. There are several areas that could benefit from strengthening:

1. Abstract: The conclusions repeat the findings- in the conclusion please describe the main take-home message. lines 47-50 can be removed

Lines 47-50 of the original manuscript have been removed and we have added the following text as a summary of the take-home message: Drawing on experiences during lockdown, VCSE organisations should develop bespoke knowledge, skills and practices to engage service users in future hybrid delivery of arts, sports, cultural and creative community activities, and to ensure that digital activities offer an equivalent degree of connection to face-to-face ones. Additionally, more effective methods of gaining feedback about patient experience of hybrid delivery is needed. 

2. Introduction. lines 82-84 are irrelevant and can be removed. Overall, the introduction could benefit from greater attention to the literature on personal recovery and the role of social prescribing. It is a bit thin and atheoretical. There is now plenty of literature on the impacts of covid-19 protective measures on mental health services that could also be referred to.

Lines 82-84 have been removed and the following text added to provide a greater basis in recovery for serious mental health and the role of social prescribing and creative endeavours:

Engagement with creative activities has also been reported to support wellbeing, help people feel happier and to promote faster recovery rates (Edmondson, Percy-Smith, Trowbridge and Walters, 2019). 

Social prescribing (SP) is one way that services link up health service provision with voluntary and community sector provision. The scope, purpose and motivation of SP is not clearly defined or agreed upon (Calderón-Larrañaga, Greenhalgh & Clinch, 2022), however, in the context of mental health recovery, a focus on person-centred design and delivery and holistic approaches that provide an environment for individualised change and development, were found to be key (Cooper et al. 2023).

3. Methods. more information is needed on sample size (target), response rate, and a description of the participants. Please describe the LEAP and their composition and function. Please also provide more information on the quantitative analysis methods. Synthesis should be described under the Analysis subheading - it does not warrant its own subheading. Under synthesis please also describe how you created the matrix reported in Table 1.

• Sample size and response rate: The additional text has been added to ‘Participants’ for clarity: 

Our aim was to get as many survey responses as possible and to conduct 8-10 interviews or as many as were needed to achieve theoretical saturation. The sample size achieved for the survey was to be all responders, which was 20 for the survey and 11 for the interviews, with an aim for 8-10 interviews to be secured from volunteers from the survey. 11 volunteers were identified, therefore interviews were increased to 11 for completeness. Responses from 20 VCSE groups represents approximately 40% of the amount of groups that were being supported by CM at the time of lockdown.

Across the 20 survey respondents, and across the interview group, the activity range was, broadly, arts and literature, horticulture and social care support. Of the 11 interviewees, 10 were female and one was male.

• Description of the participants: Descriptions of participants have not been included purposefully for reasons of anonymity. The respondents would be easily identifiable to people who attend the groups therefore it was important to maintain anonymity. 

• LEAP: Additional information has been added to Materials: The LEAP are a group of people who have lived experience of mental health services and who have an interest in research and in supporting projects to be both relevant and ethical. Participants suggested where language and content could be improved and made more understandable and accessible.

• Quantitative analysis methods: This has been changed to: The quantitative and qualitative (free text answers) were reviewed and collated and analysed as a combined data set. Quantitative data about numbers of participants lost or gained were checked against corresponding descriptive text to ensure the numbers given were actually aligned with the written descriptions given by the participants. If answers were unclear and respondents were available for interview, then further clarification was sought to ensure accuracy of reporting. Survey respondents’ were split into four groups to enable comparison between them. These groups were based on the number of additional members reached during lockdown. The data were analysed to understand the difference between providers that had not expanded their reach, those who had done so moderately, and those who had expanded to a greater degree. This comparison allowed patterns to be seen across the data. 

Results of quantitative and qualitative analyses were triangulated to obtain a richer, more complete idea of how providers adapted during lockdown and their perception of the challenges they faced. The findings from both parts of the study were compared and the research team identified where these converged, were confirmatory, and where there were discrepancies or differences.

• Synthesis title deleted. Synthesis content has been moved to follow the existing paragraph under ‘Quantitative analysis of interview data’. 

• The Mixed Methods synthesis has been expanded to clarify how the table was formulated: The matrix in Table 1 displays this information shows the extent to which the interview findings align with the data collected in the survey. Two researchers (LBE and ETB) discussed how the themes from the interviews related to the survey findings until agreement was reached for each finding. The researchers challenged each other’s decisions to ensure that full consideration had been made of each theme and the extent of the convergence/divergence.

4. Results: The figures are redundant as most information is also reported in the text. Lines 225-227 should be reported in the analysis subsection.

We thought some people might prefer visual formats however they can be removed if the editors would prefer. 

5. Discussion: The key messages and implications of the findings are somewhat unclear. The discussion could benefit from highlighting these more explicitly. Please remove all subheadings from the discussion. The list of recommendations at the end of the manuscript should be removed and either integrated into the text, where these flow from the findings being discussed or removed entirely (where they do not directly arise from the findings being reported).

The subheadings have been deleted from the discussion and it has been re-written to incorporate the suggestions made by the reviewer and to deliver a clearer summary of the implications of the findings:

This paper explored how VCSE organisations commissioned to deliver creative socially prescribed activities for people with serious mental health needs were forced to rapidly adapt their offer to comply with the COVID-19 pandemic government restrictions. We found that creative groups’ responses ranged from almost total shut down, to limited provision, to expansion to a hybrid range of offers including online, telephone contact, and at home activities. The changes made meant that learning and upskilling was rapid. This was further enhanced by listening to group member needs such as the need to feel safe online, enabling private online groups and areas, and safeguarding. These new skills were viewed as useful additions for the future for hybrid provision, though this would require funders to understand these benefits and to allow extra funds for groups to expand their offer. 

The research showed that an increase in online provision, and the use of social media, led to an increase in engaged users, though it was difficult for providers to know who those new users were, what their individual needs were and the extent to which these needs were being met. Providers who might previously have seen the creative activity as the primary mechanism of change in supporting mental health, were reminded during lockdown that connection is also an essential pillar of recovery [9]. Being able to assess the level of connection for online users, in the same way that they might assess this verbally and non-verbally for in person attendees is therefore essential for future hybrid provision. 

Additional benefits to hybrid provision of socially prescribed activities were that it did enable some people to join and connect during lockdown; a period where connection was inhibited. Post-lock this presents an opportunity for maintained connection for those who may come to experience non-Covid related isolation due to practical, physical, or mental health reasons. Examples might include people staying on hospital wards, relapse, ill health, childcare issues or people who struggle to leave their houses or meet others in in person.

We found evidence that, whilst some were enabled by online provision, such as those who would not have the access or confidence to attend in person groups, there were some groups that were less engaged, such as older adults. The range of needs encountered by the providers is significant. Recent research suggests that most people with serious mental health needs were limited users of the internet during the pandemic [10], falling on the wrong side of the pervasive social and information inequality known as the ‘digital divide’ [11]. Consequently, there is a risk that any future shift towards increased digital or online provision of socially prescribed activities will inadvertently increase social and health inequalities. The accessibility, acceptability and usability of digital provision for participants with serious mental health needs should be assessed as a matter of priority when planning hybrid groups and activities. 

Before the pandemic, providers were unlikely to routinely gather data about digital competency, learning difficulties, additional needs, or personal preferences for remote, hybrid or face to face contact. This meant that an evaluation of how lockdown impacted on accessibility and uptake in the community as a whole, as well as for specific demographic groups, was not possible. 

Social prescribing has the potential to address the social determinants of health inequalities [12]. To achieve this, provision must be accessible across the social gradient. Issues such as age, race, poverty, illness, and disability, which could increase isolation, should be at the forefront of providers minds, and demographic data on access and impact should be collected to ensure harder to reach groups are not excluded and to understand if provision is increasing or decreasing social inequalities.

Additional requirements.

The manuscript has been changed to meet the PLOS ONE style requirements. If there any additional actions to be taken, we are happy to do whatever is needed.

The following information has been added under ‘Ethics Statement’ to clarify the type of consent taken from participants.

Survey participants completed an online consent form before starting the survey. Interview participants were asked to read and complete an emailed copy of the consent form. This was returned by email to the researcher. If a participant was not able to do this verbal consent was taken at the start of the interview and recorded as part of the transcription. 

The following addition has been made to address data availability. If further contacts are required, we are happy to add as needed.

Data Availability

Due to the identifiable nature of the data it is by request only. This can be requested from The University of Sheffield Research Ethics Committee, quoting reference number 034339.

---

## [Decision Letter · Decision Letter 1]

23 Feb 2024

PONE-D-22-21627R1The delivery of creative socially prescribed activities for people with serious mental health needs during lockdown: Learning about remote, digital and hybrid deliveryPLOS ONE

Dear Dr. Bishop-Edwards,

Thank you for submitting this revised manuscript. After careful reviewing, before we can recommend acceptance, we would kindly request you to address some minor outstanding concerns regarding methodology.

Specifically, we request that you update the manuscript to provide:

a) additional information regarding the participants in the interview study (ln 138-140), in addition to the gender information already provided, such as their age range and ethnicity information, which would not compromise their individual identities

b) some examples of interview questions posed to study participants (ln 169 onwards)

The additional information requested would improve the transparency of the methodology and offer readers a deeper understanding of the responses recorded from the participants in this study. 

Please do not hesitate to reach out to us if you have any further concerns or queries regarding our revision request, and we look forward to hearing from you.

We look forward to receiving your revised manuscript.

Kind regards,

Annesha Sil, PhD

Associate Editor, PLOS ONE

Journal Requirements:

Reviewers' comments:

Reviewer's Responses to Questions

**Comments to the Author**

1. If the authors have adequately addressed your comments raised in a previous round of review and you feel that this manuscript is now acceptable for publication, you may indicate that here to bypass the “Comments to the Author” section, enter your conflict of interest statement in the “Confidential to Editor” section, and submit your "Accept" recommendation.

Reviewer #3: All comments have been addressed

2. Is the manuscript technically sound, and do the data support the conclusions?

Reviewer #3: Yes

3. Has the statistical analysis been performed appropriately and rigorously? 

Reviewer #3: Yes

4. Have the authors made all data underlying the findings in their manuscript fully available?

Reviewer #3: Yes

5. Is the manuscript presented in an intelligible fashion and written in standard English?

Reviewer #3: Yes

6. Review Comments to the Author

Reviewer #3: The article shares insight into how Voluntary Community and Social Enterprise (VCSE) organisations responded to the challenges of social distancing during the COVID-19 pandemic. The COVID-19 had multiple impacts on the human beings and overcoming it required various psycho-social measures. This article highlights some of it.

7. PLOS authors have the option to publish the peer review history of their article (what does this mean?). If published, this will include your full peer review and any attached files.

Reviewer #3: **Yes: **Syed Sajid Husain Kazmi

---

## [Author Response · Author response to Decision Letter 1]

7 Mar 2024

After careful reviewing, before we can recommend acceptance, we would kindly request you to address some minor outstanding concerns regarding methodology.

Specifically, we request that you update the manuscript to provide:

a) additional information regarding the participants in the interview study (ln 138-140), in addition to the gender information already provided, such as their age range and ethnicity information, which would not compromise their individual identities

• We do not have age or ethnicity information. 

b) some examples of interview questions posed to study participants (ln 169 onwards)

• The following examples have been added:

During the interviews participants were asked to describe: the activities they provided prior to lockdown; how they adapted or changed these activities during lockdown; which on-line platforms they used; how they stayed connected with people who did not like using on-line platforms; the challenges encountered; and which practices they intended to continue using.

---

## [Editor Report · Decision Letter 2]

18 Mar 2024

The delivery of creative socially prescribed activities for people with serious mental health needs during lockdown: Learning about remote, digital and hybrid delivery

PONE-D-22-21627R2

Dear Dr. Bishop-Edwards,

We’re pleased to inform you that your manuscript has been judged scientifically suitable for publication and will be formally accepted for publication once it meets all outstanding technical requirements.

Kind regards,

Laura Kelly

Division Editor

PLOS ONE
---

## [Editor Report · Acceptance letter]

2 May 2024

PONE-D-22-21627R2 

PLOS ONE

Dear Dr. Bishop-Edwards, 

I'm pleased to inform you that your manuscript has been deemed suitable for publication in PLOS ONE. Congratulations! Your manuscript is now being handed over to our production team.

Kind regards, 

on behalf of

Dr. Laura Hannah Kelly 

Staff Editor

PLOS ONE